# Noise Reduction Using Singular Value Decomposition with Jensen–Shannon Divergence for Coronary Computed Tomography Angiography

**DOI:** 10.3390/diagnostics13061111

**Published:** 2023-03-15

**Authors:** Ryosuke Kasai, Hideki Otsuka

**Affiliations:** Department of Medical Imaging/Nuclear Medicine, Institute of Biomedical Sciences, Tokushima University, 3-18-15 Kuramoto, Tokushima 770-8509, Japan; kasai-r@tokushima-u.ac.jp

**Keywords:** coronary computed tomography angiography, singular value decomposition, Jensen–Shannon divergence, noise reduction

## Abstract

Coronary computed tomography angiography (CCTA) is widely used due to its improvements in computed tomography (CT) diagnostic performance. Unlike other CT examinations, CCTA requires shorter rotation times of the X-ray tube, improving the temporal resolution and facilitating the imaging of the beating heart in a stationary state. However, reconstructed CT images, including those of the coronary arteries, contain insufficient X-ray photons and considerable noise. In this study, we introduce an image-processing technique for noise reduction using singular value decomposition (SVD) for CCTA images. The threshold of SVD was determined on the basis of minimization of Jensen–Shannon (JS) divergence. Experiments were performed with various numerical phantoms and varying levels of noise to reduce noise in clinical CCTA images using the determined threshold value. The numerical phantoms produced 10% higher-quality images than the conventional noise reduction method when compared on a quantitative SSIM basis. The threshold value determined by minimizing the JS–divergence was found to be useful for efficient noise reduction in actual clinical images, depending on the level of noise.

## 1. Introduction

Coronary computed tomography angiography (CCTA) has been widely used in recent years due to its the improved performance [1,2]. In particular, the scan length extension in the z-axis direction owing to the wide coverage and improvement in the X-ray tube rotation speed are the notable features of CCTA. CCTA has also been demonstrated to improve the risk assessment and further management of patients with a low-to-intermediate risk of coronary artery disease (CAD) [3,4]. However, unlike other computed tomography (CT) scans, the beating heart is the target in CCTA scans; therefore, image quality degradation due to noise can occur due to insufficient X-ray photons. Various approaches have been proposed to reduce noise. The first approach is to solve this problem by developing image reconstruction methods. Some vendors have developed iterative reconstruction methods for reducing image noise [5,6]. However, iterative reconstruction algorithms have become more algorithmically complex and require more computational power, making image reconstruction a time-consuming task.

The second approach is that the use of artificial intelligence has recently been proposed to address this problem, and the potential of deep convolutional neural networks to improve CT image reconstruction has been investigated [7]. Deep learning has been suggested to achieve this goal by minimizing the loss function between the input data and the teacher data, which are the correct data, potentially yielding significant noise reduction in comparison with conventional methods. However, deep learning has some disadvantages. It requires a large amount of training data, is difficult to implement, and has many problems that must be solved owing to the complexity of interpreting the obtained results. In particular, when the interpretation of the results is unclear, the diagnosis of medical images is greatly affected.

The third approach is to process the reconstructed image. This method is more advantageous than the above two methods because it can use previously acquired images and does not require special hardware. In addition, the simplicity of implementation and the comprehensibility of the process make it easy to interpret the results, and does not significantly affect the diagnosis of medical images. Therefore, we aimed to provide a mathematically consistent method to reduce noise in medical images, especially to focus on noise reduction in CCTA images. To achieve this goal, singular value decomposition (SVD) is the main method used in this study. SVD is used alongside principal component analysis as a method of dimensionality reduction and has been used in a wide range of fields [8,9,10]. SVD denoising takes advantage of the fact that singular values of noise are smaller than the singular values that compose the image. This is achieved by setting the number of singular values (number of ranks) to 0 for a certain threshold value. However, there is no established method of determining the threshold value. Moreover, if the threshold value is not properly determined, the image structure may be excessively corrupted or noise removal may be insufficient.

Singular values reduction method using threshold value is called low-rank approximation. The low-rank approximation is a minimization problem in which the evaluation function measures the goodness-of-fit between the given data and the approximating matrix. The appropriate evaluation function affects the determination of the threshold value and quality of the approximated image. The Frobenius norm is often used as an evaluation function to measure the level of approximation between noise-free and noisy images in low-rank approximations using SVD. This is a natural generalization of the 2-norm of a vector into a matrix. The 2-norm is known to be robust against noise; however, its convergence performance is poor, and it does not accurately capture the structure of medical images [11,12]. The thresholding of denoising by SVD requires the introduction of the effective evaluation function as a measure of the level of approximation between noise-free and noisy images. Methods using various evaluation functions have been proposed for CT image reconstruction, and their resulting high performances have been demonstrated [13,14]. Therefore, we used Jensen–Shannon (JS) divergence as an evaluation function for thresholding using low-rank approximation. It is a measure of the similarity between two probability distributions. JS–divergence is also used in generative adversarial networks (GANs), a widely used approach for generative modeling using deep neural networks [15]. When thresholding SVD using the evaluation function, it is necessary to correctly evaluate the noise-free and noisy images and provide a threshold that removes only noise. JS–divergence is expected to facilitate this process.

We evaluated the performance of the low-rank approximation through numerical experiments simulating medical images. First, various levels of noise were added to a numerical phantom that simulated a medical image, and a detailed study of the noise and singular values was conducted. The numerical phantom produced noise-free images of various noise levels and was used for thresholding to obtain the minimum JS–divergence. We also compared the results obtained by the proposed method with other noise reduction methods and confirmed the superiority of the proposed method. We then performed denoising for CCTA images from clinical CT scanner with the threshold value obtained by the numerical phantom and found that the proposed method was high performance. The advantage of using the minimum of the JS–divergence as the threshold decision is that it can be objectively determined on the basis of image quality.

## 2. Methods

### 2.1. Preliminary

The relationship between noisy and noise-free images can be described using the following equation:(1)A=e+δ
where $A\in R_{+}^{m\times n}$, $e\in R_{+}^{m\times n}$, and $\delta \in R^{m}$ denote the noisy image, noise-free image, and noise, respectively, and $R_{+}$ denotes the sets of nonnegative real numbers. The noise δ is assumed to follow a normal distribution with mean 0 and variance σ and is defined as follows:(2)δ∼N(0|σ).

Noise σ degrades the image quality and is a feature that interferes with medical image diagnosis.

### 2.2. Singular Value Decomposition

SVD was first proposed for real square matrices by Bletrami and Jordan, then extended to complex square matrices by Autonne, and finally to general rectangular matrices by Eckart and Young [16]. SVD is also a popular method of matrix factorization in linear algebra, and a special case of eigenvalue decomposition can be formulated as follows:(3)$$A=U\Lambda V^{⊤}$$
where $U\in R_{+}^{m\times m}$ is an orthogonal matrix consisting of the left singular vector of *A*. $V\in R_{+}^{n\times n}$ is an orthogonal matrix consisting of the right singular vector and ⊤ denotes the transpose. $\Lambda \in R_{+}^{m\times n}$ is a diagonal matrix with singular values, $\lambda_{i}\geq \lambda_{2}\cdots \geq \lambda_{n}\geq 0$. Let *i* denote the number of nonzero singular values of *A* or, equivalently, the rank of *A* (where i=rank(A)≤n).

Equation (Equation 3) can be rewritten as follows:(4)$$A=\sum_{i=1}^{n}u_{i}\lambda_{i}v_{i}^{⊤}.$$

Equation (Equation 4) is used to describe the denoised estimated matrix *A* by using a low-rank matrix [17].

### 2.3. Jensen–Shannon Divergence

KL–divergence is widely known as divergence [18]. Divergence is a function that coincides with the Riemannian distance if the distance is microscopic on the manifold. Let *p* and *q* be two non-negative vectors.
(5)$$\mathrm{KL}\left(p,q\right)=\sum_{i=1}^{n}p_{i}\log\frac{p_{i}}{q_{i}}+q_{i}-p_{i}.$$

Although the KL–divergence measure is sometimes referred to as a metric in the probability distribution, the KL–divergence is not symmetric and does not strictly satisfy the axiom of distance as a metric. In general,
(6)KL(p,q)≠KL(q,p).

Specifically, it does not satisfy the triangular inequality, and a practical problem of indeterminate values exists when there is a region where p≠0,q=0. Minimizing the KL–divergence has an important property in that it is equivalent to maximizing the likelihood function modeled by the probability distribution.

The JS–divergence was introduced by Lin as a measure of the discrepancy between two or more discrete probability distributions to overcome these limitations [19].

The generalized JS–divergence is defined as follows:(7)$$\begin{array}{ccc} \mathrm{JS}(p,q) & = & \frac{1}{2}\mathrm{KL}\left(p,\frac{1}{2}\left(p,q\right)\right)+\frac{1}{2}\mathrm{KL}\left(q,\frac{1}{2}\left(p,q\right)\right)\\ & = & \frac{1}{2}\sum_{i=1}^{n}p_{i}\log\frac{p_{i}}{\frac{1}{2}\left(p_{i}+q_{i}\right)}+\frac{1}{2}\sum_{i=1}^{n}q_{i}\log\frac{q_{i}}{\frac{1}{2}\left(p_{i}+q_{i}\right)}. \end{array}$$

The JS–divergence symmetrizes the KL–divergence by taking the average relative entropy of the source distributions to the entropy of the average distribution [20,21,22,23]. JS–divergence, which also exhibits high performance in the domain of image recognition, has the following properties in addition to symmetry [24,25,26]. It takes non-negative elements and is zero if and only if all the elements are identical. The square root of itself is a metric. In particular, the fact that the elements are nonnegative is particularly suitable in the image domain.

### 2.4. SVD with JS–Divergence

Noise reduction by SVD is performed by selecting an appropriate number of ranks *k* to estimate *e* in Equation (Equation 1). Various methods have been proposed for selecting the number of the rank [27,28,29,30]. However, in this study, the number of ranks that minimized the JS–divergence between the noise-free image and the noisy image was selected as the optimal number of ranks for denoising. Therefore, we had set the following evaluation function for the optimal rank selection of SVD based on our proposed minimization of JS–divergence:(8)$$E\left(A\left(i\right)\right):=\mathrm{JS}\left(e,A(i)\right),$$
and the optimal rank *k* approximation of a denoising matrix *A* under JS–divergence can be formulated as follows:(9)$$A=\underset{s.t.\mathrm{rank}(A)=k}{\mathrm{argmin}}E\left(A\left(i\right)\right).$$

The rank *k* that yields the minimum value in Equation (Equation 8) is set as the threshold value, and the singular value after the threshold value is set to 0 to remove noise from the image. By using JS–divergence as an evaluation function, we expected to eliminate high-performance noise in low-rank approximations of images while preserving the diagnostically important structure of medical images.

## 3. Experimental Results and Discussion

In this section, We will conduct experiments to validate our proposed method. First, we show the results for a numerical phantom. Next, we provide the results of comparison with other noise reduction methods. Finally, we confirm and discuss the results of using the proposed method on CCTA images obtained from a clinical CT scanner.

### 3.1. Numerical Phantom

The usefulness of the proposed method was verified using a numerical phantom for which the true pixel value *e* is known. In medical imaging, it is difficult to obtain a noise-free image using a clinical CT scanner. In this study, a noise-free image was used as the true image and the correct data were obtained using a numerical phantom. The numerical phantoms used in our study were the Shepp–Logan phantom, which is popular for evaluating performance in CT image reconstruction [31], and a simulated water phantom consisting of cylindrical rods of different sizes placed in an acrylic container in water, as illustrated in Figure 1a,b.

Each numerical phantom had a true noise-free pixel value e∈[0,255] and was composed of 512 × 512 pixels. The entire process from phantom creation to numerical computation was performed using Python 3.10.9. To reproduce the noise usually observed in CT images for both the Shepp–Logan and the water phantoms, the noise of the normal distribution in Equations (Equation 1) and (Equation 2) was added with variances of 15, 20, and 25. Images with noise added to the true image in each phantom are shown in Figure 2.

First, we verified the singular value $\lambda_{i}$ of the Shepp–Logan and water phantoms in Equation (Equation 4). The plot of the singular value for each phantom is shown in Figure 3.

The graph shows plots of noise-free and various noise levels, and the singular values are normalized to a maximum value of 1. The vertical axis shows the singular value, with the maximum value restricted for ease of observation while the horizontal axis represents the value of *i*. The singular values rapidly decreased as the number of rank in the image matrix increased. For higher orders, the values are close to 0. It is clear from each of the graphs that the singular value shifts from the true image *e* to larger values due to noise. Additionally, the graph also clearly demonstrates that the higher the noise level, the larger the singular value. Furthermore, it was difficult to determine the index at which the singular value should be a threshold from these graphs. Separating noise from the important components of an image required a threshold method different from the graph for a singular value.

Next, we observed the singular vector for a singular value. Figure 4 shows the value of singular vectors in Equation (Equation 4) for the noise-free image *e* and for σ = 20, the middle of each noise level. An example using the Shepp–Logan phantom. Orange dot groups shows a noise-free image, and green dot groups shows σ = 20. All graphs were drawn on the same scale, with horizontal line segments in the graphs representing the zero level. From the graphs, the singular vectors are concave and convex with short periods as the value of *i* increases, that is, their spatial frequency increases. However, when noise is added, there is no periodic structure, and the graph is a scattering graph indicating randomness. Thus, the value of *i*, when thresholded by a singular value, dominates the trade-off between the fine structure and noise elements of the image. While the noise in the image can be suppressed for small values of *i*, higher frequencies are also removed, making it impossible to represent fine structures. In CCTA images, resolution reduction may affect the peripheral vascularization of coronary arteries, thus making the threshold value an important factor in determining image resolution.

We examined the threshold results using the JS–divergence proposed in this study as the evaluation function. A graph plotting the value of the evaluation function $log_{10}\left(E\left(A\left(i\right)\right)\right)$ for each *i* between the noise-free image *e* as a true image and the noisy image is shown in the upper panels of Figure 5. The vertical axis value of the evaluation function denotes the logarithmic value $log_{10}\left(E\left(A\left(i\right)\right)\right)$. The evaluation function E(A(i)) is 0 when the distance between the two distributions is equal, i.e., when the low-rank approximate image and noise-free image *e* are equal. A lower value indicates a closer match to the true image. The lower panel shows the position of the minimum value of the evaluation function for each level of noise, as indicated by the red dots and vertical bars. The value of the index *i* at the time of the minimum value of the evaluation function is the optimal threshold value *k* in SVD.

The true image, which is noise-free, and the image containing noise are measured by the evaluation function E(A(i)) based on the JS–divergence, yielding the image closest to the true image. The minimum value of each evaluation function $log_{10}\left(E\left(A\left(k\right)\right)\right)$ and the index value i=k are summarized in Table 1.

As the graph shows, as the level of noise increases, the value of threshold *k* tends to decrease, and the noise reduction becomes stronger. This trend was also observed in for the Shepp–Logan and water phantoms.

Figure 6 and Figure 7 show the optimal threshold *k* value that yields the minimum value of the evaluation function and the obtained noise-reduction image. The top row shows the true image and its density profile, and the red line in the image indicates the position of the acquired density profile. Next to the noise-reduced image for each noise level σ = 15, 20, and 25, the original image with noise and the concentration profile of the reduced image are shown. Although the randomness of the density profile increases with increasing noise level for both the Shepp–Logan and the water phantoms, the noise-reduced image obtained with the proposed method shows that the randomness can be suppressed at any noise level. In particular, the density profile of the noise σ = 25 of the water phantom in Figure 7 shows a low contrast, but the density profile of the noise-reduced image can discriminate contrast differences. By obtaining the density profile, evaluations such as visual and quantitative evaluation are possible and can be performed.

To further confirm the effectiveness of our proposed method, we compared it with other noise reduction methods. To solve the problem of noise reduction in medical images, various methods are used in practice. We compared the wavelet transform, which is a typical noise reduction method [32,33]. Figure 8 and Figure 9 show the comparison of the proposed method and wavelet transform denoising with the images and its density profiles. Shepp–Logan and water phantoms indicate that the proposed method is effective in reducing noise. In particular, at the high noise level of σ = 25, we confirmed that the proposed method effectively achieves noise reduction, whereas the wavelet transform is not sufficient to remove noise. From the viewpoint of quantitative evaluation, the structural similarity index measure [34] (SSIM) between the noise reduction and noise-free images was calculated in Table 2. The SSIM is a perception-based quality index, and higher values of SSIM indicate higher image quality. The SSIM values also show that the proposed method produces high values for all noise levels, indicating that the images are of high quality.

### 3.2. SVD Using CCTA

We aimed to validate the proposed method on a numerical phantom by using CCTA images obtained from a clinical CT scanner. Unlike numerical phantoms, noise-free images are difficult to obtain in clinical imaging. However, using the results obtained from previous numerical phantoms, noise reduction can be applied to clinical images. Figure 10 shows images of CCTA obtained from the clinical CT scanner.

CCTA images were acquired from a 320-row CT Aquilion ONE Vision Edition (Canon Medical Systems, Tochigi, Japan) by ECG-synchronized scanning. The scan conditions were as follows: tube voltage, 120 kV; tube current, 600 mA; and rotation time, 0.275 s. The entire heart was scanned once as a volumetric scan. The thickness and spacing of the reconstructed slices were 0.5 mm. Image reconstruction was performed using half-reconstruction. As seen from Figure 10, of the CCTA image, the rotation time of the X-ray tube must be made shorter to obtain a stationary image. This resulted in a lack of X-ray photons and a reconstructed image is noisy. Therefore, we used our proposed noise reduction method for CCTA images. From numerical phantom experiments, the index giving the minimum value of the evaluation function was used as the threshold for noise reduction. The problem of not being able to obtain noise-free images in a clinical CT scanner was solved by using the results of numerical phantom experiments.

First, to determine the threshold in the clinical image, the region of interest (ROI) shown in Figure 11 was set and the level of noise was measured.

Although it is difficult to estimate the amount of noise directly from clinical images, the simplest method is to obtain the standard deviation from the ROI. The standard deviation in this CCTA image was σ = 19.8. Therefore, the standard deviation was almost the same as that of the noise N(0|20) added to the numerical phantom. From Table 1, *k* = 66, which is the lowest value of the evaluation function when σ = 20, which was set as the threshold value in CCTA. Figure 12 shows the results of noise reduction with *k* = 66. The density profiles are shown with normalized scaling, such that the maximum value is at 1 and the minimum value is at 0. As with the numerical phantom, the red line shows the profiles before and after noise reduction. As can be seen from the images and profiles, the randomness owing to noise was reduced. The results obtained from the numerical phantom were used to determine the threshold value for noise reduction, and as with the numerical phantom, noise reduction with a high level of accuracy was achieved.

Another advantage of noise reduction using SVD is that it can reduce the dimensionality of the image because it removes singular values from the image by thresholding, whereas noise reduction using ordinary pixel-based filtering cannot reduce the dimensionality of the image. Reducing the dimensionality of features by removing noise, which is an additional component in machine learning, is advantageous for building an efficient learning system. The thresholding method based on JS–divergence proposed in this study enables noise removal while preserving the important components of the image.

## 4. Conclusions

We proposed a new thresholding method for noise reduction by SVD using JS–divergence as an evaluation function. A noise-free image cannot be obtained in clinical imaging. However, by using a numerical phantom, noise-free images were obtained for use in thresholding based on the minimum value of JS–divergence. JS–divergence functioned as an evaluation function to separate important structures from noise in the image and provided high-quality images. Furthermore, by comparing the threshold value obtained from the numerical phantoms with the noise levels of clinical CCTA images, we justified that this approach for noise reduction is also effective for clinical images. By using JS–divergence, a well-known tool for comparing probability distributions, as an evaluation function, it is possible to compare pixels as probability distributions. By mapping comparisons between distributions to noise-free and noisy images, the minimum value of JS–divergence can provide the threshold that yields the best image, as opposed to SVD, for which the determination of the threshold has traditionally been empirical. In the future, we plan to develop a new method to reduce noise in more complex structures of numerical phantoms and use clinical images other than CCTA. Furthermore, it is necessary to examine how the noise and dimensionality reduction functions of SVD affect the results and diagnosis as training data for machine learning in future studies.

## Figures and Tables

**Figure 1 diagnostics-13-01111-f001:**
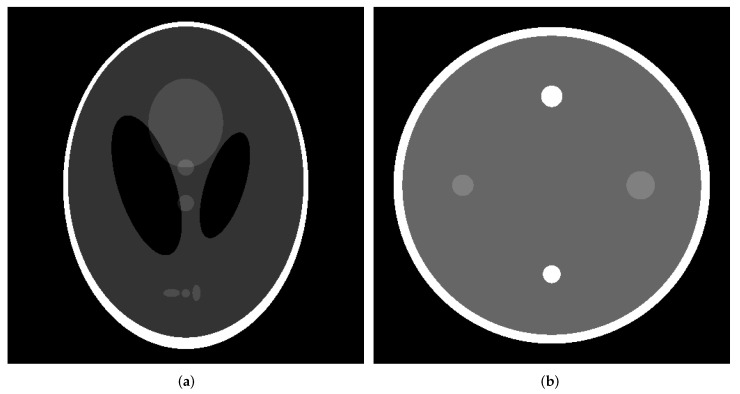
Numerical phantom. (**a**) Shepp–Logan phantom. (**b**) Water phantom.

**Figure 2 diagnostics-13-01111-f002:**
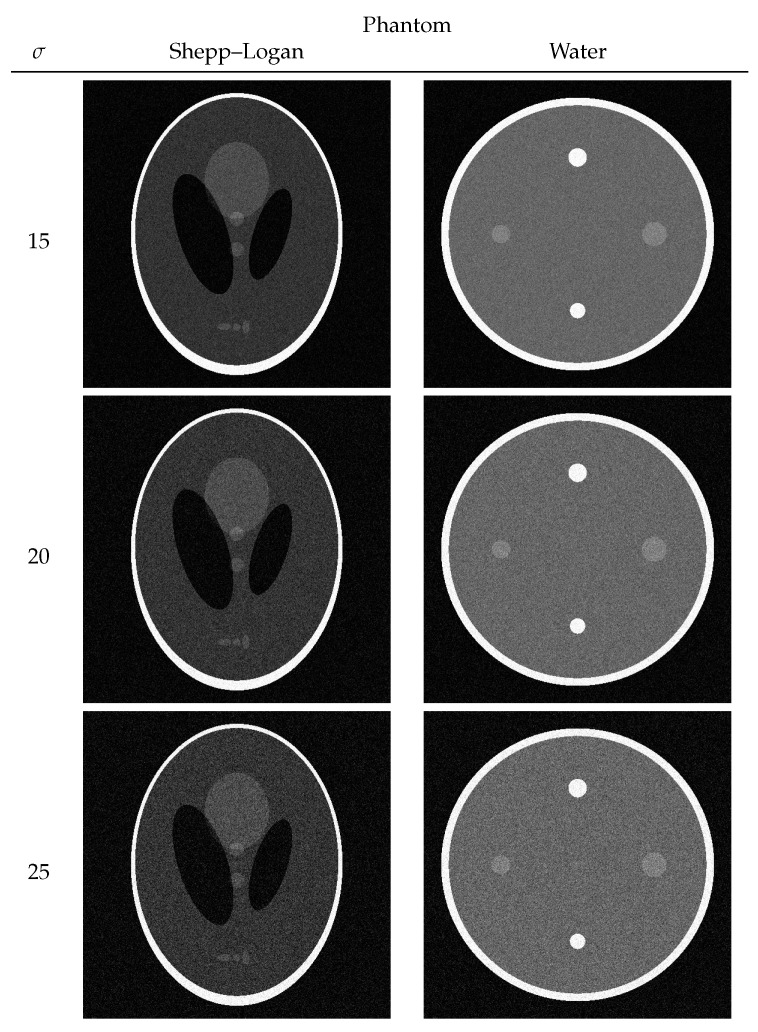
Shepp–Logan and water phantoms with noise N(0|σ).

**Figure 3 diagnostics-13-01111-f003:**
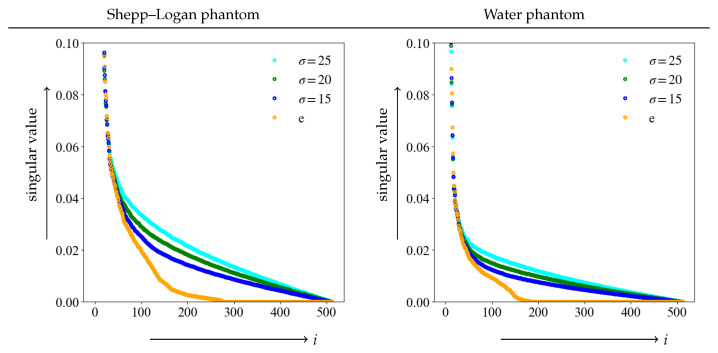
Graph of singular value in the true image *e* and the respective noise level. The maximum singular value is set to 1, and is restricted for easy observation.

**Figure 4 diagnostics-13-01111-f004:**
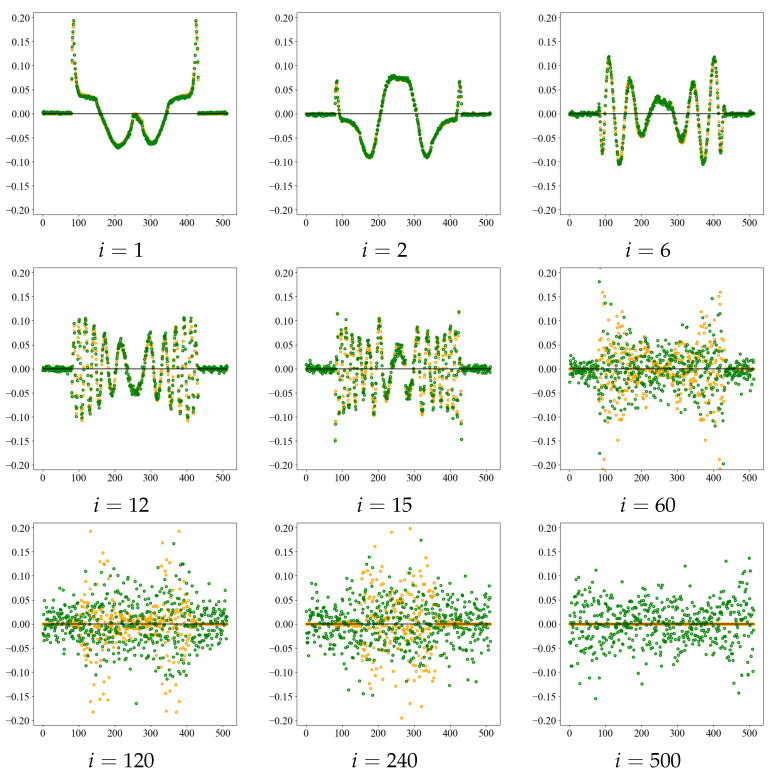
Singular vectors at the noise-free images *e* and noise σ = 20 at each rank number *i*. Orange dot groups shows the noise-free image and green dot groups shows σ = 20 as an example of noise level.

**Figure 5 diagnostics-13-01111-f005:**
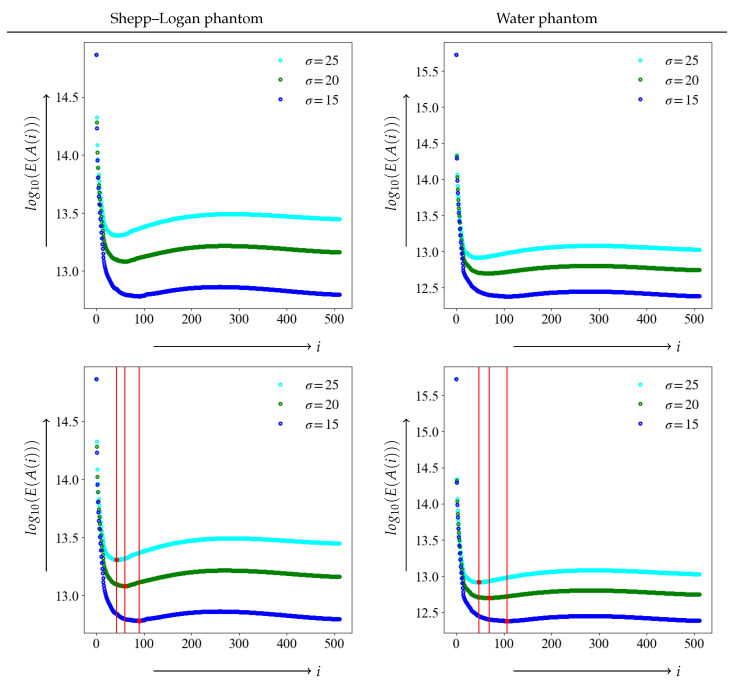
Upper panel corresponds to graphs of JS–divergence $log_{10}\left(E\left(A\left(i\right)\right)\right)$ at noise σ = 15, 20, and 25. The lower panel depicts the minimum value i=k and the position of the threshold with dots and vertical bars.

**Figure 6 diagnostics-13-01111-f006:**
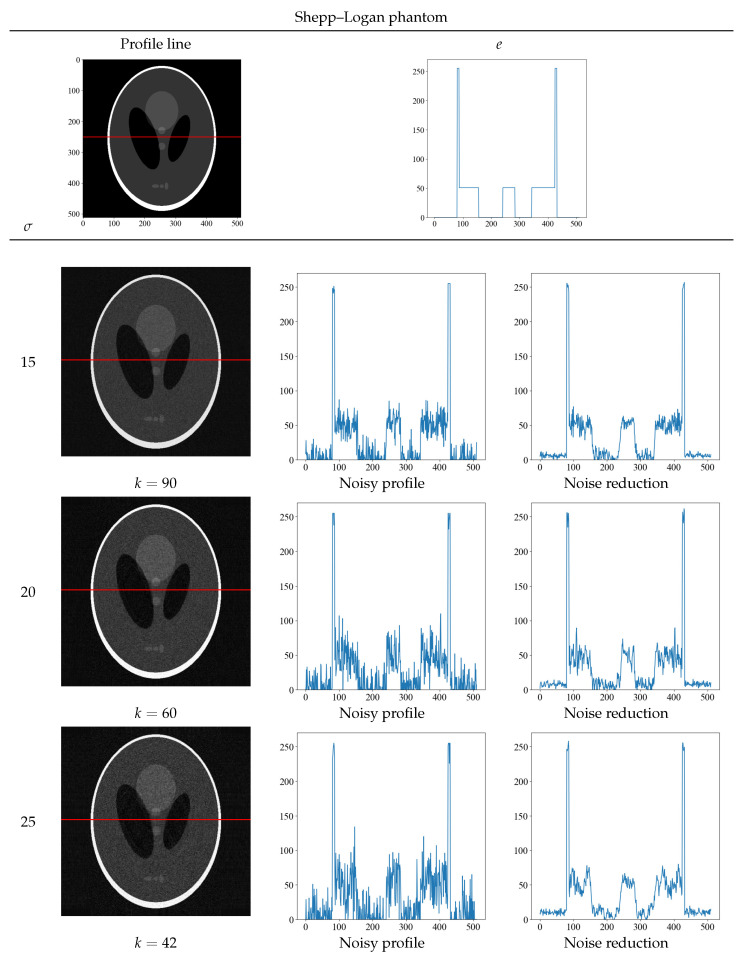
The optimal threshold *k* that yields the minimum value of the evaluation function at each noise level and the resultant noise-reduced image of the Shepp–Logan phantom. The top row shows the true image and its density profile. The red line depicts the area where the density profile was obtained. The profiles below the top row show the concentration profiles of the noisy and noise-reduced image.

**Figure 7 diagnostics-13-01111-f007:**
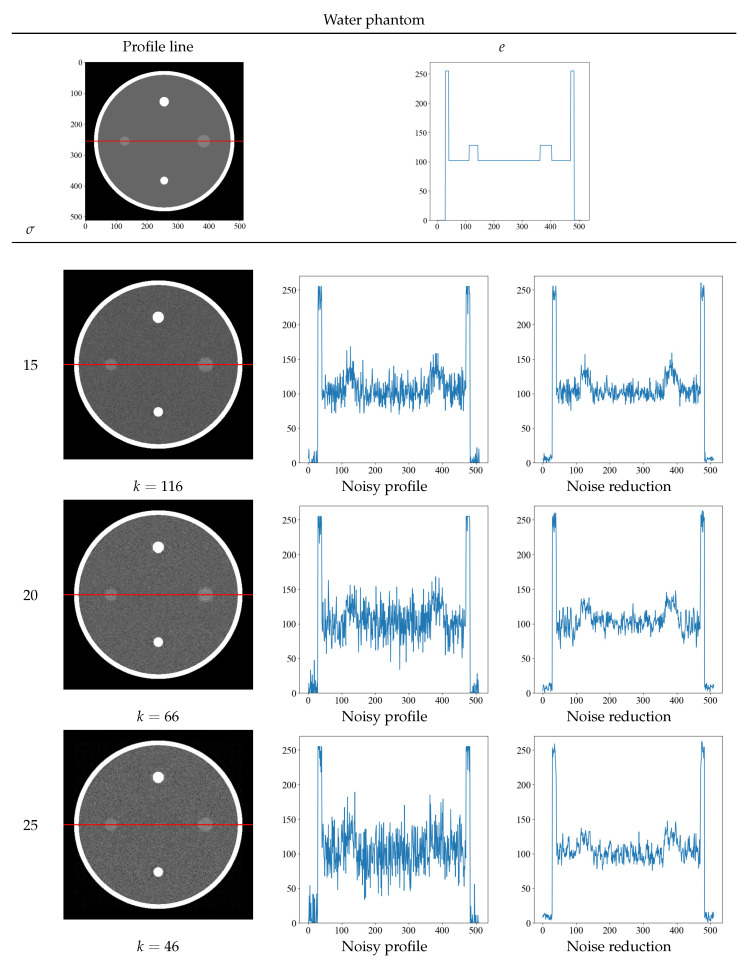
The optimal threshold *k* that yields the minimum value of the evaluation function at each noise level and the resultant noise-reduced image of the water phantom. The top row shows the true image and its density profile. The red line depicts the area where the density profile was obtained. The profiles below the top row show the concentration profiles of the noisy and noise-reduced image.

**Figure 8 diagnostics-13-01111-f008:**
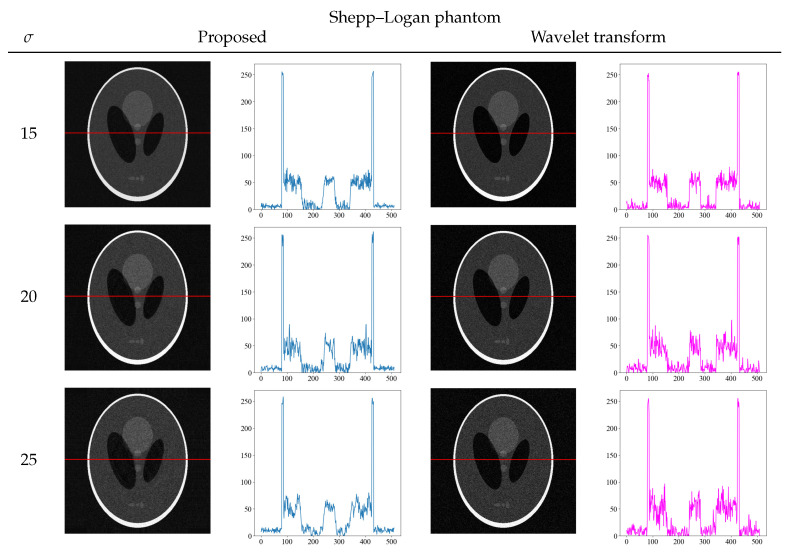
Comparison images and density profiles between the proposed method and noise reduction by wavelet transform.

**Figure 9 diagnostics-13-01111-f009:**
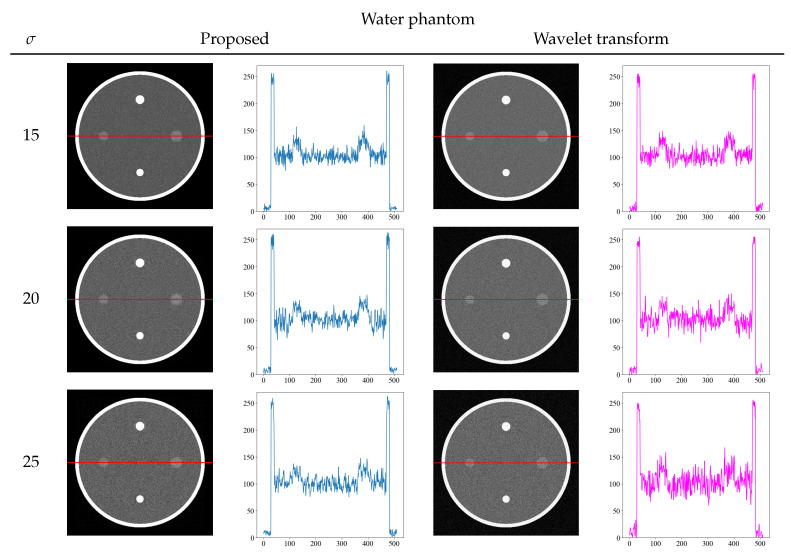
Comparison images and density profiles between the proposed method and noise reduction by wavelet transform.

**Figure 10 diagnostics-13-01111-f010:**
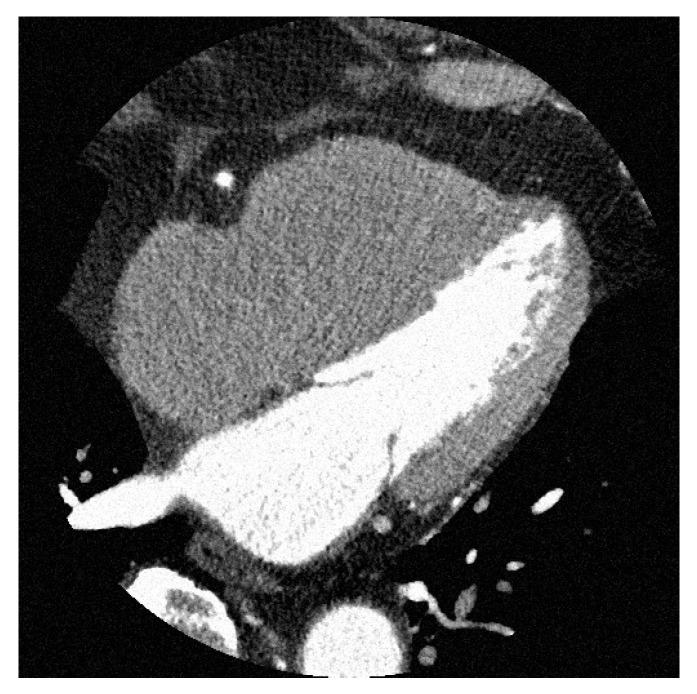
Axial image of CCTA acquired from a clinical CT scanner. It can be seen that the image contains a significant amount of noise.

**Figure 11 diagnostics-13-01111-f011:**
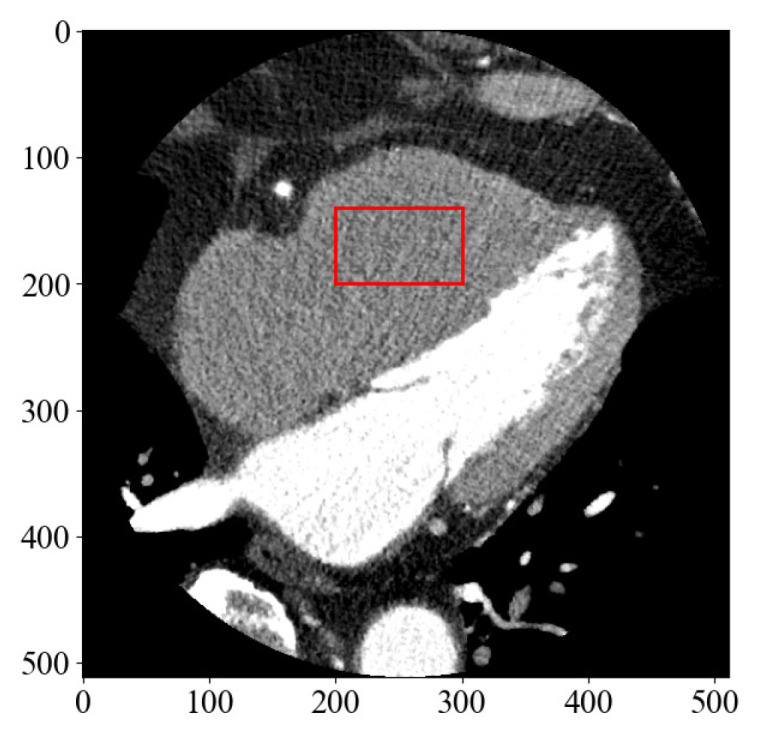
A CCTA image displaying the location of the region of interest (ROI) to measure the level of noise in the image.

**Figure 12 diagnostics-13-01111-f012:**
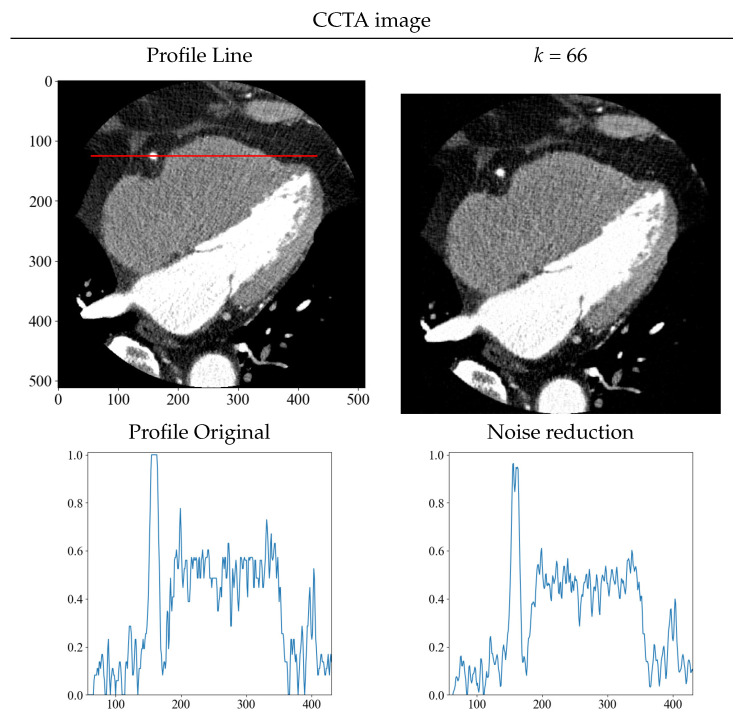
Original and noise-reduced image, and their respective density profiles before and after noise reduction, are displayed. The red line depicts the area where the density profiles were obtained on the image before noise reduction.

**Table 1 diagnostics-13-01111-t001:** $log_{10}\left(E\left(A\left(k\right)\right)\right)$ minimum of JS–divergence for noise σ = 15, 20, and 25 in each phantom and threshold i=k.

σ	Shepp–Logan Phantom		Water Phantom	
	$\log_{10}\left(E\left(A\left(k\right)\right)\right)$	k	$\log_{10}\left(E\left(A\left(k\right)\right)\right)$	k
15	12.7817	90	12.3518	116
20	13.0832	60	12.6728	66
25	13.3065	42	12.8896	46

**Table 2 diagnostics-13-01111-t002:** SSIM for noise reduction by the proposed method and wavelet transform.

σ	SSIM			
	Proposed		Wavelet Transform	
	Shepp–Logan Phantom	Water Phantom	Shepp–Logan Phantom	Water Phantom
15	0.7331	0.7047	0.6781	0.6298
20	0.7130	0.6826	0.6431	0.5556
25	0.6968	0.6589	0.6026	0.5198

## Data Availability

All data used to support the findings of this study are included within the article.

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
