# Peer review of "Noise Reduction Using Singular Value Decomposition with Jensen–Shannon Divergence for Coronary Computed Tomography Angiography"

_diagnostics, 2023, doi:10.3390/diagnostics13061111_

Round 1

Reviewer 1 Report

The manuscript entitled “ Noise reduction using singular value decomposition with Jensen–Shannon divergence for coronary computed tomography angiography “has been investigated in detail. The topic addressed in the manuscript is potentially interesting and the manuscript contains some practical meanings, however, there are some issues which should be addressed by the authors:

·         The problem statement is not explained comprehensively and lack background study. Background study needs improvements.

·         There are no numerical results in the Abstarct section. Please add results to this section.

·         It will be helpful to the readers if some discussions about insight of the main results are added as Remarks.

·         Please set the figures 6-10 before than conclusion section.

·         Please add a comparison table to the manuscript and compare your method with other noise reduction methods such as wavelet transform.

This study may be consider for publication if it is addressed in the specified problems.

Reviewer 2 Report

Noise reduction is a challenge in signal/image processing.

Please explain why Figures 6 to 10 from pages 10 to 14 must be put between "4. Conclusions" without any detailed information provided.

Round 2

Reviewer 1 Report

My recommendation is accept in current form.